# Research Progress on Microstructure Evolution and Strengthening-Toughening Mechanism of Mg Alloys by Extrusion

**DOI:** 10.3390/ma16103791

**Published:** 2023-05-17

**Authors:** Yaqi Zheng, Yuan Zhang, Yun Liu, Yaqiang Tian, Xiaoping Zheng, Liansheng Chen

**Affiliations:** Key Laboratory of the Ministry of Education for Modern Metallurgy Technology, College of Metallurgy and Energy, North China University of Science and Technology, Tangshan 063210, China

**Keywords:** extrusion, microstructure evolution, toughening mechanism, DRX, texture

## Abstract

Magnesium and magnesium-based alloys are widely used in the transportation, aerospace and military industries because they are lightweight, have good specific strength, a high specific damping capacity, excellent electromagnetic shielding properties and controllable degradation. However, traditional as-cast magnesium alloys have many defects. Their mechanical and corrosion properties cause difficulties in meeting application requirements. Therefore, extrusion processes are often used to eliminate the structural defects of magnesium alloys, and to improve strength and toughness synergy as well as corrosion resistance. This paper comprehensively summarizes the characteristics of extrusion processes, elaborates on the evolution law of microstructure, discusses DRX nucleation, texture weakening and abnormal texture behavior, discusses the influence of extrusion parameters on alloy properties, and systematically analyzes the properties of extruded magnesium alloys. The strengthening mechanism is comprehensively summarized, the non-basal plane slip, texture weakening and randomization laws are comprehensively summarized, and the future research direction of high-performance extruded magnesium alloys is prospected.

## 1. Introduction

Magnesium alloys have been widely applied in various fields, including engineering, transportation, aerospace and biomedical industries, owing to their low density (1.7 to 2.0 g/cm^3^), high specific strength, superior damping performance, excellent biosafety, complete degradability and recyclability [1]. However, the deformation of Mg alloys restricts their applications, owing to a hexagonal close-packed (HCP) structure which has insufficient independent slip systems. There has been a great deal of research on promoting the structure density, reducing the internal defects, and optimizing the mechanical properties and corrosive behavior of as-cast magnesium alloys by extrusion deformation processes [2,3,4,5,6,7]. Extrusion forming technologies can maximize the plastic potential of Mg alloys by applying three-dimensional compressive stress which can effectively refine grains. Guo et al. [4] found that a hot extrusion process can significantly weaken the basal texture intensity of Mg-5Li-3Sn-2Al-1Zn alloy and improve elongation without loss of tensile yield strength (YS) or ultimate tensile strength (UTS).

As is well known, extrusion processes refer to those advanced forming processes in which high strains are imposed on original bulk samples, leading to grain refinement. With the development of technology, many extrusion processes have been proposed and researched. Commonly used methods include direct extrusion, indirect extrusion [8], equal channel angular pressing [9] and rotary extrusion [10]. Among these different extrusion methods, there are attractive methods owing to their relatively simple procedures, reduced material handling and large deformations. Regardless of several advantages that extrusion methods have, unfortunately, there are also limitations and disadvantages, including uneven distribution of strain, heterogeneous microstructure and performance, and a requirement for high stress and complex mold structure [11].

Compared with other magnesium alloys deformation processes, such as die-casting, rolling and forging, extrusion processing is a simple, low-cost, one-time process of magnesium alloys such as plates and bars, which can effectively improve the deformation capacity of the alloy and expand the application range of magnesium alloys. During the extrusion process, Mg alloy is deformed under a three-way compressive stress, and the microstructure changes significantly. Dynamic recrystallization (DRX) during extrusion, static recrystallization (SRX) after extrusion, the change of grain orientation, grain refinement, dynamic precipitation process of the second phase and the crushing and redistribution of large-size eutectic phase are the main factors which influence the mechanical and corrosive properties of magnesium alloys. The microstructure evolution and mechanical properties of Mg alloys mainly depend on the composition and extrusion conditions, including extrusion ratio (ER), speed and temperature. The mechanical properties of magnesium alloys decrease with the increase in grain size caused by increasing extrusion temperature, and the effect of ER on the alloy properties is more complicated. Feng et al. [12] explored the extrusion behavior of Mg-8Li-3Al-2Zn-0.5Y (wt.%) alloy at 200 °C and found that with the ER changed from 4:1 to 25:1, grains were refined and stretched. The tensile properties first increased and then decreased. The alloy exhibited the highest yield strength (YS, 214 MPa), UTS (243 MPa) and elongation (EL, 41%) when the ER was 16. Go et al. [13] found that the tensile yield strength of extruded Mg-5Bi-9Al alloy was 223 MPa. Moreover, the highest TYS of extruded Mg-14Gd-0.5Zr-1Sm alloy with a higher rare earth element content can reach 374 MPa, and the EL was 8.8% [14]. The ultimate tensile yield strength of the as-extruded Mg-15Gd-2Nd alloy exceeded 250 MPa and the elongation exceeded 10% [15]. The YS and EL of Mg-Gd-Er-Zr alloy prepared using a double-pass extrusion method can reach 422 ± 1.59 MPa and 6.5 ± 0.89% [16].

Because the extrusion process greatly improves the performance of magnesium alloys, exploring the essence of its performance improvement can optimize the process and clarify the best process parameters for use in the future. Therefore, this paper describes the microstructure evolution of Mg alloys in detail and comprehensively summarizes the influence of extrusion parameters on the microstructure evolution and properties of magnesium alloys, such as strengthening mechanism, corrosion resistance and corrosion fatigue behaviors. Moreover, elaborate dynamic recrystallization and texture evolution are described. This work can provide guidance for extruded Mg alloys.

## 2. Magnesium Alloy Extrusion Process and Advantages

Most magnesium alloys, with high strength-to-weight ratio and low density, are lightweight structure materials and attract attention for various applications. At present, the processing of magnesium alloys mainly involves hot rolling which always needs to be formed through a multi-pass rolling process. Compared with the traditional multi-directional forging and multi-pass forging processes, the extrusion processes are simpler and can achieve higher grain refinement. In addition, the extrusion process can provide a three-way compressive stress, which can effectively prevent the cracking caused by poor plastic formability of Mg alloys, and the required sheet, pipe or bar can be obtained by one pass extrusion. In addition, hot rolling, forging and other processes cannot obtain many products with special cross-section shapes. With regard to small cross-section rods, it would be advisable to mention the methods of continuous extrusion, in which the lack of discreteness is eliminated. Extrusion can greatly improve the mechanical and corrosion properties of the alloy while optimizing the process. At present, extrusion technologies can mainly be divided into direct extrusion, indirect extrusion, equal channel angular extrusion and rotary extrusion.

### 2.1. Direct Extrusion

Direct extrusion is a process in which the sample outflow direction is the same direction as the extrusion axis. During the extrusion process, the extrusion rod pushes the as-cast Mg alloy under heating into the extrusion cylinder. Under the action of the main plunger, the extrusion shaft forces the magnesium alloy in the extrusion cylinder to flow out through the die hole, and finally the ingot moves forward slowly. Direct extrusion has many advantages, such as a flexible process, the ability to create diverse product shapes, a simple equipment structure, easy mold accessibility and simple production operations (Figure 1) [17,18]. However, due to the movement of Mg alloys in the extrusion cylinder, a large friction force is generated which causes metal to flow unevenly and produces many geometric wastes during the extrusion process. Moreover, the extrusion force of direct extrusion is 30% to 40% larger than that of indirect extrusion. Compared with aluminum alloys, Mg alloys are easy to bond with the die during extrusion and have a large frictional adhesion. A uniform temperature is not easy to achieve when Mg alloys are heated. Therefore, a uniform flow could be obtained under the conditions of ensuring uniform temperature and sufficient lubrication during the extrusion process. In addition, grains in the center region are larger than those at the edge, a hardened layer can easily grow on the surface of samples, the mechanical properties of the sample change greatly from edge to center, and the formability of samples are poor.

Huang et al. [19] found that the UTS, YS and EL of extruded Mg-Al-Ca-Mn alloy increased to 270 MPa, 195 MPa and 15.9%, respectively. The co-segregation of Ca and Al increased the cohesive energy of grain boundaries to promote the mechanical properties and inhabited intergranular fracture to improve the elongation. Moreover, the refined second phase, strengthened grain boundaries and modified texture are beneficial to improve the mechanical strength and plasticity. Wang et al. [18] proved that hot extrusion improved the strength and EL of Mg-9Gd-5Y-0.5Zr alloy due to the formation a <0001> basal texture. The extruded Mg-2Zn-1.6Ca achieved an outstanding room temperature ductility of 18.08% with a reasonable reduction in mechanical strength [20].

### 2.2. Indirect Extrusion

Compared with direct extrusion, indirect extrusion can obtain samples with more uniform deformation in cross-section and length and can process large products. The production is efficient and the continuity of production can be efficiently improved. Backward extrusion adds an additional shear stress, resulting in a significant refinement of microstructure in just one cycle. Under the same extrusion pass, the structure of indirect extrusion is more uniform, and the mechanical properties are better [8]. Mg wheels are always formed using an indirect extrusion process (Figure 2a).

Wang et al. [21] described the effect of accumulative alternating indirect extrusion on the mechanical properties of commercial AZ31 alloy and stated that after accumulative alternating indirect extrusion, the grain of the extrusion sheet was significantly refined. With the number of extrusion cycles increased from 2:1 to 4:1, dislocations with a high density increased the proportion of DRX. Then the grains became finer, the texture became weak, and the material plasticity increased from 23.99% to 38.48% (TD direction). Cheng et al. [8] evaluated the relationship between microstructure evolution and extrusion conditions of backward-extruded Mg-6Sn alloy and found that the texture strength decreased with an increasing extrusion speed and temperature. As the extrusion temperature decreased, the grain size became smaller. Fatemi-Varzaneh et al. [22] found that the average grain size of extruded AZ31 alloy could reach 1 μm.

**Figure 2 materials-16-03791-f002:**
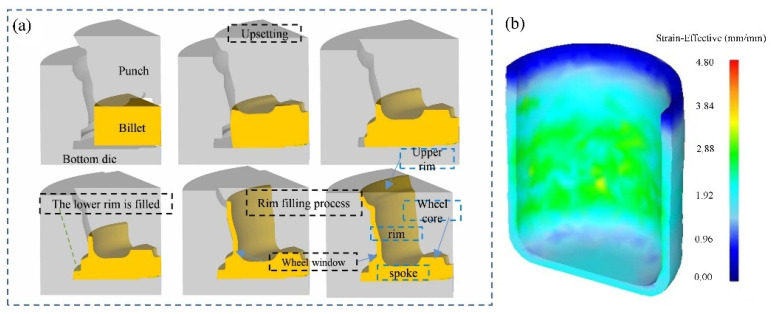
Schematic diagram of indirect extrusion [23] and distribution of the effective strain for indirect extrusion processed sample [24]: (**a**) indirect forming process (grey color) of a Mg alloy wheel (brown color); (**b**) the distribution of the effective strain for an indirect extrusion processed sample.

### 2.3. Equal Channel Angular Pressing

The die of equal channel angular pressing (ECAP) is mainly composed of two equal diameter channels with a certain angle, and the corner could shear stress that is not available in direct extrusion (Figure 3 [25]). Due to the size of the sample does not change significantly, ECAP can perform muti-pass repeated accumulation deformation, which can greatly improve the grain refinement limit. ECAP is one of the severe plastic deformation (SPD) processes used for preparing ultra-fine-grained and ultra-high-strength magnesium alloy due to its simple mold, controllable strain and high grain refinement. The equivalent strain can be estimated according to Equation (1) [26]:(1)$$\varepsilon_{eq}=\frac{N}{\sqrt{3}}\left[2\cot\left(\frac{\Phi +\psi }{2}\right)+\psi csc\left(\frac{\Phi +\psi }{2}\right)\right]$$
where Φ describes the channel angle, *Ψ* signifies the outer curvature of the mold, and *N* is the equal channel angle extrusion pass.

Liu et al. [9] confirmed that the mechanical properties of Mg-4Li alloy after four-pass ECAP were relatively stable and the grains were alternately elongated or spheroidized. The mechanical strength of Mg-4Li-3(AL-12Si)-1(Al-5Ti-1B) alloy reaches the peak value after two passes of ECAP (UTS, YS and elongation can reach 243 ± 2 MPa, 170 ± 2 MPa and 16.4 ± 0.4%). High-pass ECAP will lead to the growth of α-Mg, and the agglomeration and dissolution of AlLi particles, resulting in decreased mechanical properties of the alloy. Sun et al. [27] revealed that 32-pass equal channel angle pressing can synergistically enhance the mechanical and the corrosive properties of Mg-Al-Ca-Mn. This was due to the uniform and refined grains (about 2.6 μm), the rapid build-up of the corrosion film and the enhancement of the (0001) basal texture density.

### 2.4. Rotary Extrusion

Rotary extrusion is a process which combines a traditional extrusion process with torsional deformation. The cumulative strain of the blank is obviously increased during rotary extrusion deformation (Figure 4 [28]). The flow stress is obviously lower than that of indirect extrusion and the equivalent stress increases with the reduction in rotation. The sum of the equivalent of torsion and indirect extrusion can be described using Equation (2) [29]:(2)$$\in_{T}=\ln\left(\frac{2\pi N\hat{A}\cdot \gamma \hat{A}\cdot h_{0}}{h^{2}}\right)+\ln\frac{R_{0}^{2}}{R_{0}^{2}-R_{1}^{2}}$$
where *N* signifies the number of revolutions, *h* and *h*_0_ represent the final and initial bottom thickness of the specimen, and *R*_0_ and *R*_1_ are the punch radius and the die inner radius. The shear stress (τ) and shear strain (γ) can be converted by torque-rotation, and the expression [10,30] is:(3)$$\tau =\frac{M}{2\pi R^{3}}(3+n+m)$$
(4)$$\gamma =\frac{R\theta }{L}$$
where *R* corresponds to the punch radius, θ signifies the torsion angle, L describes the extrusion length, *n* stands for the strain hardening coefficient, and *m* signifies the strain rate sensitivity factor (*n* = *m* = 0).

With certain torsion process parameters, the torsion extrusion specimen shows gradient change. Yu et al. [10] found that the flow stress (100 MPa) of the torsional extrusion process of Mg-Gd-Y-Zn-Zr was significantly lower than that of indirect extrusion (180 MPa). Texture orientation gradually weakened along the direction of the radius. After indirect extrusion, the (0001) basal orientation coexisted with $\left[10\bar{1}0\right]$ and $\left[2\bar{1}\bar{1}0\right]$ textures. At the same time, Yu et al. [10] found that the microhardness of Mg-11.95Gd-4.5Y-2Zn-0.37Zr (wt.%) alloy decreased with an increase in torsional revolutions, and the matrix and the second phase were more uniformly distributed. The grain refinement performance has a positive effect on the rotational speed and the DRX refinement is in dynamic competition with growth at a constant temperature and strain rate. Li et al. [31] proposed another rotary extrusion process named friction stir extrusion and increased the YS, UTS and compression strain of rare-containing Mg alloys by 42.5%, 63.6% and 35.5%, respectively. Shear assisted processing and extrusion (ShAPE) could work with more than 20 times less ram force compared to conventional extrusion [32]. The reduction is mainly due to the rotational shear component applied by ShAPE interacting with a scrolled die face, which promotes the flow of Mg alloy into the die orifice. Rotating backward extrusion could effectively increase the equivalent strain and deformation uniformity of the Mg alloys more than conventional indirect extrusion [33].

## 3. Effect of Extrusion on Mg Alloys

The extrusion process can significantly change the microstructure evolution, texture formation and mechanical properties of magnesium alloys. Compared to original Mg alloys, the grain size of extruded Mg alloys is significantly decreased. The extrusion temperature has a significant effect on grain size. The extruded microstructure was mostly equiaxed grains at the extruded temperature of between 200 °C and 400 °C, indicating that it had a relatively obvious dynamic recrystallization behavior. With the increasing extrusion temperature, grain size gradually increased, and then the grain boundary diffusion and migration behaviors were intense. In addition, the ER also has an obvious influence on microstructure evolution. In general, texture strength decreased with an increase in extrusion ratio and a formation of $<\bar{1}2\bar{1}1>$ texture parallel to the ED. Mg-Zn-Yb-Zr alloy formed a $<04\bar{4}3>$-$<\bar{1}2\bar{1}1>$ texture with a maximum intensity of 1.5 m.u.d. at 320 °C (ER of 20:1) [34]. Xiao et al. [35] found that with an increase in ER, the texture intensity of Mg-Zn-Gd-Zr decreased and the deviation angle of the basal plane, the extrusion direction and the ratio of YS to TYS (TCYA) increased. The evolution of the texture, including a reduction in the texture and a shift in orientation from the basal texture, improved the effect of the extrusion ratio on TCYA.

### 3.1. Mechanical Properties

The extrusion process is a powerful way to improve defects such as shrinkage and porosity in as-cast alloys. It can significantly refine and uniform the microstructure of the alloy so that the mechanical properties can meet the requirements of practical applications. Extrusion process parameters affect the mechanical properties of alloys. In general, increasing the extrusion ratio can refine the grains, and the UTS and YS of the alloy are improved. The extrusion temperature determines the degree of recrystallization. As the temperature increases, the degree of recrystallization increases. Under certain extrusion parameters, the mechanical properties of Mg alloy are directly related to the texture. Extruded alloys exhibit high tensile strength along the extrusion direction (ED) [36,37].

Ma et al. [38] stated that the UTS, TYS and EL of Mg-1.5Gd alloy could reach 195 MPa, 135 MP and 42% at an extrusion temperature of 360 °C. The TS and YS decreased first and then tended to be stable with an increase in extrusion temperature. A maximum EL (49.3) and YS (84 MPa) of Mg-1.5Gd alloy were acquired at 420 °C. Luo et al. [39] found that the UTS, TYS and EL of Mg-10Gd-1Zn-0.4Zr alloy could reach 257 MPa, 166 MPa and 15.6% at 500 °C. Whereas, at a low extrusion temperature of 460 °C, the UTS, TYS and EL were improved to 301 MPa, 219 MPa and 23.9%. This is because the Schmid factor of the extrusion alloy with an extrusion temperature of 460 °C of {0001} was much larger than that of 500 °C. The alloy is strengthened by using the following equation:(5)$$\sigma_{\mathrm{gb}+t}=\frac{0.3}{SF}\left(\sigma_{0}+kd^{-1/2}\right)$$
where *σ*_gb+t_ is the total contribution of texture strengthening and grain boundary strengthening, *k* is the Hall–Petch constant of magnesium alloy containing LPSO phase (188 MPa μm^1/2^) and *d* and *σ*_0_ are the average and original lattice strength (11 MPa) of pure magnesium, respectively.

Kiani et al. [40] found that the UTS and EL of extruded Mg-0.035Zr-0.2Sr alloy were 254 MPa and 11.1%. Xu et al. [41] stated that the best mechanical properties of Mg-0.035Zr-0.2Sr alloy were obtained at an extrusion temperature of 320 °C due to the combined interaction of grain boundary, dislocation and precipitation strengthening. The YS, UTS and EL could be 258.5 MPa, 283.8 MPa and 19.1%. Kim et al. [42] compared the mechanical properties of AZ91 and AZ91-0.3Ca-0.2Y at different extrusion temperatures and found that the alloy showed finer grains, lower texture strength, and higher YS at 300 °C than when extruded at 400 °C. Zhou et al. [43] coordinated the high strength of Mg-Li alloy with an Mn element addition and hot extrusion deformation. The YS, UTS and EL of extruded Mg-Li-Al-Mn alloy could reach 248 MPa, 332 MPa and 14.3%. This high strength was obtained by the synergistic action of grain boundary, precipitation and dislocation strengthening. Compared with the conventionally extruded Mg-5Li-2Zn alloy [44] (YS, UTS and YL were 195 MPa, 116 MPa and 17%), the YS and UTS of Mg-4Li-3Al-0.3Mn was greatly improved, whereas the EL was slightly reduced. The tensile properties of some extruded magnesium alloys are shown in Table 1. It is obvious that the mechanical properties of extruded Mg alloys are excellent.

### 3.2. Corrosion Performance

The extrusion process cannot only effectively improve the mechanical properties of Mg alloys, but it can also affect corrosive behavior. Corrosion resistance is affected by the redistribution of solution in the microstructure evolution, grain refinement, and the change in texture orientation and strength [81].

Peng et al. [82] revealed the influence of extrusion on the corrosion resistance of homogenized Mg-Li-Al-Zn-Y alloys. It was found that the corrosion properties of extruded alloys were preferable to those of homogenized alloys. This was because the extrusion process made the second phase distribution of Mg-Li-Al-Zn-Y alloy more uniform, obtained a relatively complete oxide film by sacrificing the β-Li phase to protect the matrix and decreased the corrosion rate. Xu et al. [41] described the effect of the extrusion process on the in vitro degradation of Mg-Zn-Gd-Mn-Sr alloy. It was found that the extruded alloy had relatively uniform corrosion and the lowest corrosion rate at an extrusion temperature of 360 °C. The microstructure (extruded at 360 °C) changed from a bimodal structure to a uniform structure composed of equiaxed grains. The volume proportion of the W phase decreased, the volume fraction of HAGBs increased, and uniform intergranular corrosion and pitting corrosion behavior occurred on the Mg matrix. Bazhenov et al. [83] found that the extrusion temperature had little influence on the corrosion rate of Mg-4Zn-0.7Ca alloy. The corrosion rates of the extruded alloy were 0.97 mm/y (200 °C) and 1.18 mm/y (300 °C). Considering the confidence limit, there was no essential difference in the corrosion rate of these alloys. This can be attributed to the subtle difference in grain size and phase composition. Luo et al. [84] discovered that the smaller the grain size of the Mg-6Gd-2Y-0.2Zr alloy was, the more easily the alloy corroded. Furthermore, the (0001) plane has the highest corrosion resistance for low surface energy (15.4 kJ/mol). Yamasak et al. [85] tested the corrosion resistance of Mg-Zn-Y-Al alloys on vertical and cross sections and found that the extrusion process caused different grain orientations, in which the vertical section was related to the <1010> axis and cross section was mainly related to the {1010} plane. This was the major reason for the difference between the corrosion rate of the vertical (0.7 mm/y) and cross sections (0.35 mm/y) in 3.5 wt.% NaCl solution. Song et al. [86] also found that the (0001) basal plane in extruded AZ31 had a higher chemical stability and corrosion resistance than $\left(10\bar{1}0\right)$ and $\left(11\bar{2}0\right)$ crystal planes. Lotgpour et al. [87] found that the increase in grain boundaries during extrusion would change the corrosion resistance of Mg-2Zn-0.3Cu alloy. The corrosion rate decreased from 10.5 ± 0.2 to 7.5 ± 0.2 mm/y. Grain boundary is of great significance to corrosion propagation and passivation layer formation. Due to the excellent passivation effect of extruded alloy, a uniform passive layer was produced on the surface which hinders the corrosion process. Baek et al. [88] found that the corrosion rate (27.5 mm/y) of Mg-8Sn-1Zn-1Al alloy can be effectively reduced by using the indirect extrusion process with an extrusion speed of 1.3 mm/s and at an ER of 25. The low-temperature extrusion process had an obvious improvement on the corrosion resistance of Mg-8Sn-1Zn-1Al alloy. The SKPFM spectrum (Figure 5) shows the electrochemical potential of Al_5_(Fe, Mn)_2_. With a decrease in extrusion temperature from 280 °C to 180 °C, the solid solubility of Mn was almost unchanged, whereas the diffusion of Fe element was slow at low temperature, so Mn can be densely segregated in Al_5_(Fe, Mn)_2_ particles. The substitution of Mn for Fe improved the thermodynamic stability of Al-Fe phase, because the decrease in Fe/Mn ratio will decrease the potential of the second phase and improve the overall corrosion resistance.

Wu et al. [89] tested the corrosion properties of as-cast and extruded Mg-2Ca-*n*Al (*n* = 0, 2, 3, 5 wt.%) alloys and found that the corrosion resistance of the alloy changed from 20.8 ± 5.9 mm/y, 17.1 ± 0.9 mm/y, 5.9 ± 2.4 and 9.6 ± 2.9 mm/y to 26.1 ± 4.6 mm/y, 2.7 ± 0.4 mm/y, 4.9 ± 1.3 mm/y and 7.4 ± 1.6 mm/y. The formation of more network corrosion channels deteriorated the corrosion resistance of Mg-2Ca alloy, whereas the increase in corrosion resistance of Mg-2Ca alloy with 2, 3, 5 wt.% Al composition was attributed to more grain boundaries generated by grain refinement, which provided nucleation sites for the passivation film and which could also be used as a physical corrosion barrier. Compared with a coarse grain structure, the corrosion rate of alloys with a smaller grain structure was slower. Xu et al. [90] found that the extrusion ratio could affect the corrosion resistance of Mg-Y alloy. With an increase in extrusion ratio, the change rate of corrosion potential became relatively larger, and the cathodic corrosion current moved in a positive direction, indicating that the corrosion resistance improved with the increase in ER. The arc radius of the vertical section was slightly larger than that of the cross section, which meant that the corrosion rate of the vertical section was smaller. Orlov et al. [91] proved that an integrated process of extrusion and ECAP can reduce the corrosion rate of ZK60 alloy. Compared with the corresponding ED surface, the decrease of corrosion current density on the ND-ED surface under SPD treatment can be attributed to the change in crystal structure. Corrosion pits were more likely to nucleate on the {0001} plane and the propagation speed along the prismatic $\left(10\bar{1}0\right)$ was faster. After SPD treatment, the longitudinal section of ZK60 was mainly composed of (0001) basal plane and {11-20} crystal plane. The ND-TD surface mainly contained $\left\{10\bar{1}0\right\}$ plane, so the Nd-TD corrosion rate was faster.

### 3.3. Corrosion Fatigue Performance

For biomedical magnesium alloys, corrosion fatigue is a key indicator used to evaluate the overall performance of the alloy under fatigue-environment interaction. Compared with as-cast alloys, the extrusion process can obviously prolong the corrosion fatigue life of magnesium alloy in NaCl solution and other corrosive environments.

Gu et al. [92] explored the corrosion fatigue properties of extruded WE43 and found that the fatigue limit of WE43 was reduced from 110 MPa (10^7^ in air) to 40 MPa (10^7^ in SBF), which was about one order of magnitude higher than that of as-cast AZ91D alloy. Corrosion fatigue life is related to microstructure, environment and cyclic load. The fracture mode of WE alloy was ductile fracture, and a corrosion fatigue crack was initiated from the micropore defects in the sample. Jafari et al. [93] found that the grain boundary strengthening of extruded ZX10 alloy and the absence of cathode IMP can effectively strengthen corrosion fatigue resistance. The basal texture of the extruded alloy at 325 °C and 400 °C led to the c-axis of the majority of the grains to be perpendicular to the ED. Therefore, the activation of tensile twins had a very effective Schmid factor during the compression process, and the process of detwinning the structure may have occurred during the stretching process. The residual twins were the result of competition between compression twinning and tensile detwinning. However, the twins were the initiation point of a corrosion fatigue crack, which provided a potential path for corrosion fatigue crack growth and accelerated corrosion fatigue fracture.

Jiang et al. [94] explored the corrosion fatigue behavior of extruded Mg-0.5Zn-0.2Ca alloy in NaCl solution and found that the corrosion fatigue limit in deionized water (ATW) and deionized water (DIW)-based 0.5 wt.% NaCl solution was 115 MPa and 105 MPa. A crack initiated from inclusions in the NaCl solution and hydrogen embrittlement, which were the main factors, and anodic dissolution was the minor factor in fatigue crack growth. During corrosion fatigue tests, cyclic loading and/or the local development of a surface layer with poor corrosion protection can lead to oxide film damage which facilitates the entry of hydrogen. Therefore, local hydrogen accumulation will be formed, resulting in crack propagation. On the other hand, the weak protective layer meant the matrix of Mg alloy dissolved more, thereby reducing corrosion fatigue resistance. The extruded Mg-0.5Zn-0.2Ca had better fatigue resistance in ATW-based solution than in DIW-based solution, and the reason for this should be the enhancement of its anti-dissolution ability. The corrosion fatigue behavior of extruded AZ31 alloy in 3 wt.% NaCl solution could be divided into two processes: the corrosion pit propagation stage and the crack propagation stage before failure. In the initial stage of the corrosion fatigue process, the corrosion pit was formed by the preferential corrosion dissolution of the second phase and grew under cyclic stress. A corrosion crack originated at the bottom of the corrosion pit when the depth of the corrosion pit reached the critical value. In the low-stress region, corrosion cracks accounted for 70–80% of the fatigue life before initiation. Therefore, the behavior of corrosion pit growth determined the corrosion fatigue life of extruded Mg alloys in the low-stress region.

## 4. Deformation Mechanism of Extrusion

Extrusion temperature is one of the major factors affecting the plastic deformation mechanism of Mg alloys. With an increase in temperature, the CRSS of the non-basal slip system including prismatic slip and cone slip decreases and can easily reach the activation value during high-temperature extrusion. Moreover, twinning deformation is also the main deformation mechanism for Mg alloys. On the one hand, twinning competes with slip. On the other hand, twinning and slip can work together to adapt to deformation.

### 4.1. Slip Deformation

The {0001}$<10\bar{1}1>$ basal slip is the major deformation mechanism because of its low CRSS at room temperature. Generally, basal slip is more easily activated due to its low CRSS during low-temperature extrusion. Wang et al. [95] revealed the extrusion deformation behavior of AZ61 alloy at 200 °C and found that temperature has little influence on the CRSS value of twins but has a significant effect on slip. No obvious $\left\{10\bar{1}2\right\}$ tensile twins are found in Figure 6, and the contribution of twins is weak. The Schmid factor of the $\{11\bar{2}2\}<\bar{1}\bar{1}23>$ second-order pyramidal slip system is large, representing that the deformation is mainly due to the $\{11\bar{2}2\}<\bar{1}\bar{1}23>$ secondary order pyramidal slip.

### 4.2. Twinning Deformation

The deformation of Mg alloys is also influenced by twins, and the common deformation mechanisms include $\left\{10\bar{1}2\right\}$ tensile twins and $\left\{10\bar{1}1\right\}$ compressive twins. The occurrence of twin shear displacement is usually smaller than that of sliding deformation. However, twins can weaken the original matrix and effectively accommodate strain deformation [96]. At present, the nucleation and growth mechanism of twins in Mg alloys can be described by the Schmid factor and the characteristic of adjacent grains, which are controlled by grain morphology, orientation and size [97]. However, there are few studies on twins in extruded alloys, and a quantitative relationship between twin characteristics and extrusion parameters is not clear.

Sabbaghian et al. [98] found that the three surface structures of extruded Mg-4Zn alloy were composed of uniform equiaxed grains with different content of twins. There were almost no twins on the ED surface, whereas TD and 45° surfaces contained a large number of twins (volume fractions of 1.1%, 14.6% and 5.6%, respectively). Although the contribution of twins to plastic formation was limited, twins can act as an additional barrier to dislocation movement, so TD samples with a higher number of twins showed higher strength during deformation. Wang et al. [99] also found that the number of twins in different planes was different. The longitudinal cross-section surface (LS) plane had a lower density of twins than the transverse cross-section surface (TS). The LS and TS planes were mainly composed of $\left\{10\bar{1}0\right\}$ and $\left\{10\bar{1}2\right\}$$<11\bar{2}0>$ plane, whereas the twins were mainly $\left\{10\bar{1}0\right\}$-$\left\{10\bar{2}0\right\}$ twins and $\left\{10\bar{1}2\right\}$ twins. Kiani et al. [40] stated that the microstructure of extruded Mg-Zr-Sr alloy was mainly constituted of recrystallized grains, coarse grains and twins. This meant that no DRX occurred in these coarse grains, and the deformation was dominated by twins. In subsequent deformation, compressive loading in an ED direction led to deformation through twins, resulting in low σ_CYS_ and high σ_UCS_.

## 5. Strengthening Mechanism of Extrusion

### 5.1. Fine Grain Strengthening and Precipitation Strengthening in DRX Process

Hot extrusion can usually obtain a fine recrystallized microstructure and then obviously promote the mechanical properties of Mg alloys [39]. Generally, extrusion conditions (extrusion temperature, speed and ratio) can determine the grain refinement and recrystallization behavior. Magnesium alloys with high alloying content will dynamically precipitate a large number of fine, second-phase particles during extrusion, which will affect the DRX behavior of the alloy through its grain boundary pinning effect [100,101,102]. In addition to dynamically formed precipitates, the original microstructure of the magnesium alloys before extrusion also affects DRX behavior during the extrusion process, for instance, smaller original grains provide more high-angle grain boundaries and more points for DRX nucleation and promote DRX behavior during extrusion [103]. The extrusion parameters also affect DRX behavior, for instance, small extrusion ratio and low extrusion temperature can lead to incomplete recrystallization behavior, which can promote strength at the expense of ductility.

Luo et al. [39] found that the precipitation of Mg_5_Gd phase from α-Mg matrix and the fragmentation of bulk LPSO were two main PSN mechanisms which can promote DRX during hot deformation. Sadi et al. [104] stated that the DRX mechanism of extruded Mg-0.3Ce (wt.%) alloy under low strain was mainly twinning-induced dynamic recrystallization (TDRX) and particle-stimulated nucleation (PSN). With the increase in strain, DRX was dominated by sub-grain development (SD) and a grain boundary bulging dynamic recrystallization (GBDRX) mechanism. In Figure 7, new grains with different orientations generate from the deformed grains, and the dislocations forming sub-grains also accumulate at the position shown by the arrow in (c).

Kim et al. [105] reported on the DRX behavior of Mg-7Sn-1Al-1Zn alloy at an extrusion temperature of 250 °C and 450 °C. It was found that a large number of nanoscale Mg_2_Sn precipitates were formed which led to the formation of uniformed DRX grains, delaying DRX during the extrusion process at low temperature. The cooling process of extrusion also affects DRX behavior. Wu et al. [106] compared the DRX behavior of air-cooling and force-air-cooling extruded Mg-Gd-Y-Zn-Zr alloy and found that the DRX ratios of air-cooling and force-air-cooling samples were 92% and 73%, respectively. Continuous DRX started at the original coarse grain boundaries and developed towards the interior of the coarse grain during extrusion. The distribution spacing of the layered LPSO phase and the dynamically precipitated β phase in force-air-cooled alloy was closer than that of air-cooled alloy which can effectively fix the DRX grain boundaries and limit the grain growth. The grain size was refined from 2.9 to 1.4 μm (Figure 8). Li et al. [107] believed that during the secondary extrusion process of Mg-Gd-Y-Zr alloy, DRX occurred in the shear band, resulting in randomization of the texture, formation of a weak texture aligned with $<10\bar{1}0>$∥ED and changing from $<10\bar{1}0>$ to $<11\bar{2}0>$ pole. During the extrusion process, the intragranular precipitates were arranged linearly or dispersedly along the triple junction of the grain boundary. At room temperature, plenty of fine precipitates and solute atoms could strengthen the grain boundary and hinder the movement of dislocations in the crystal. Grain strengthening was the major reason for the strengthening of the extruded alloy.

### 5.2. Texture Strengthening

In addition to the DRX behavior during hot extrusion, texture is another important factor affecting mechanical properties of extruded magnesium alloys. In general, alloying elements and extrusion conditions are two important factors affecting the texture of magnesium alloys. Strong basal texture perpendicular to ED is formed in rare-earth-free extruded magnesium alloys. Owing to the addition of rare earth elements, the solute drag effect and pinning effect of rare-earth-rich phases are promoted. The extruded Mg alloy will form a texture parallel to ED, and the basal texture becomes weak, which is advantageous to basal slip and improves the ductility of Mg alloy [106,108,109,110,111]. Texture can affect the mechanical properties according to the Hall–Petch constant:(6)$$\sigma_{0}=m\tau_{0}$$
where *m* represents the orientation factor which is determined by the texture strength. σ_0_ is the Hall–Petch line intercept constant, *τ*_0_ stands for the CRSS of the basal slip. 

Borkar et al. [112] found that the basal texture of extruded Mg-Mn alloys was parallel to ED. The addition of Sr can weaken the basal texture of Mg-Mn alloys by PSN. More Sr was needed to weaken the texture during a low-speed extrusion. The addition of RE helps to promote ductility. In the extruded Mg-Gd alloy, recrystallized grains and elongated grains mainly had a $<10\bar{1}0>$ orientation parallel to ED [113]. The preferred growth of the $<2\bar{1}\bar{1}1>$ grain and abnormal grain formation led to the strengthening of the $<2\bar{1}\bar{1}1>$ texture component. The difference in stored energy between the $<2\bar{1}\bar{1}1>$ grain with a low KAM and the $<10\bar{1}0>$ grains with a high KAM provided the driving force for grain growth during recrystallization. The high-angle grain boundaries between $<2\bar{1}\bar{1}1>$ grains and $<10\bar{1}0>$ grains had a higher mobility than the low-angle grain boundaries between $<2\bar{1}\bar{1}1>$ grains, so the grains grew more easily. Wang et al. [114] stated that the presence of abnormal texture increased the σ_0_ and k in the Hall–Petch formula, leading to an increase in yield strength. Wu et al. [115] found that a double extrusion process increased the strength of Mg-8Gd-4Y-1Nd-0.5Zr alloy (362 MPa) more than single extrusion (428 MPa). In the secondary extrusion, it was easier to break the coarse phase and promote dissolution in the matrix. The abnormal texture appeared after the secondary extrusion, which meant that a $<10\bar{1}0>$ texture parallel to ED direction was transformed to $<11\bar{2}1>$ parallel to ED direction. The rare earth elements also have a disordered and weakening effect on extrusion texture [108]. The texture of Mg-Gd-Y-Zr alloy after secondary extrusion was weaker than that of the single extrusion, but it was still the basal texture.

The extrusion condition can also affect the texture strength of magnesium alloys. Jin et al. [116] proved that the abnormal <0001> extrusion texture//ED in extruded Mg-Gd-Y-Zn-Zr alloy was the result of non-substrate slip continuous operation. The strength of abnormal texture increased with the increase in extrusion temperature and ER and demonstrated that recrystallization cannot promote the growth of abnormal texture in subsequent annealing treatment, whereas DRX contributed to its formation. With an increase in deformation temperature and strain, non-basal dislocation will move easier, and the continuous accumulation of dislocation from c Burgers vector to sub-grains will cause sub-grains to rotate to <0001>//ED, leading to an abnormal extrusion texture. Gneiger et al. [117] tested the microstructure of Mg-Ca-Si alloy at extrusion temperatures of 300 °C and 350 °C and found that texture strength decreased from 11.8 m.r.u. to 4.5 m.r.u. This was because the alloy formed a bimodal structure composed of DRX and non-DRX regions when extruded at 300 °C, and a completely recrystallized structure (93%) was formed at 350 °C. The non-DRX regions showed higher texture strength. Bi et al. [118] also found that different elements also affected the texture strength. The maximum texture strengths of extruded MYZ, MYZN and MYN were 6.34, 10.13 and 5.63 m.r.u., respectively.

## 6. Conclusions

In this paper, the research progress concerning the contribution of extrusion to the microstructure, mechanical properties, corrosion behavior and corrosion fatigue properties of magnesium alloys in the past decade were reviewed, including the relationship between extrusion parameters and microstructure evolution, mechanical properties and corrosion behavior.

The extrusion process can efficiently refine the grain of Mg alloy and improve its mechanical properties via the grain boundary strengthening mechanism. Moreover, the DRX behavior and texture of Mg alloys during extrusion can significantly improve the properties of alloys. Rare earth Mg alloys generate rare earth texture in the extrusion process, weakening the basal texture strength, and this texture is conducive to the occurrence of basal slip and elongation increase.The asymmetry caused by texture hinders the application of Mg alloys in key fields, so it is important to study the relationship between process and texture intensity, non-basal dislocation activation and twin formation.The deformation mechanism of magnesium alloy during extrusion is not clear, and the relationships between the dominant position of slip, twins and extrusion conditions needs to be established. In addition, the contribution of slip and twins to deformation has not been evaluated and quantitatively calculated.Due to the significant enhancement effect of the extrusion process on magnesium alloys, the extrusion pressure should be reduced in future by combining the optimization of the extrusion process with the setting of die parameters.Although the extrusion process enhances the comprehensive mechanical properties and corrosion resistance of alloys, the microstructure of the alloy is still insufficient, which is far from reaching the limit of the properties of magnesium alloy. Energy-field-assisted magnesium alloy extrusion forming, which may involve ultrasound, lasers, or magnetism, could optimize microstructure and promote manufacturing and is one of the potential development directions.

## Figures and Tables

**Figure 1 materials-16-03791-f001:**
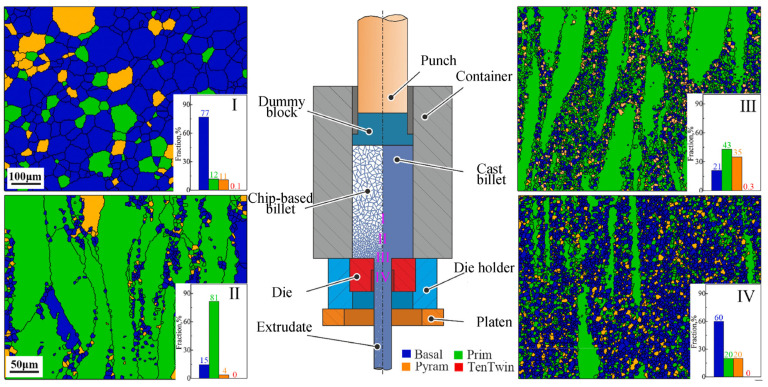
Schematic diagram of direct extrusion [17] and microstructure in stage I to IV [18].

**Figure 3 materials-16-03791-f003:**
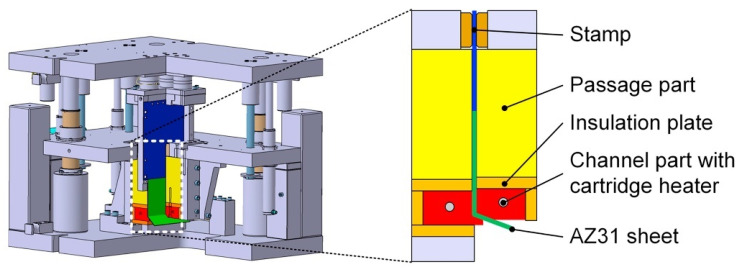
Schematic diagram of ECAP [25].

**Figure 4 materials-16-03791-f004:**
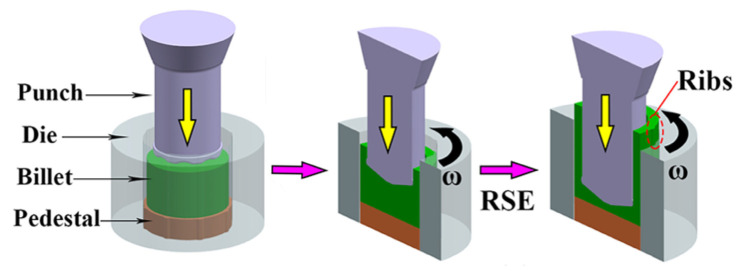
Schematic of rotary extrusion process with a pressure (yellow arrow) torsional force (black arrow) [28].

**Figure 5 materials-16-03791-f005:**
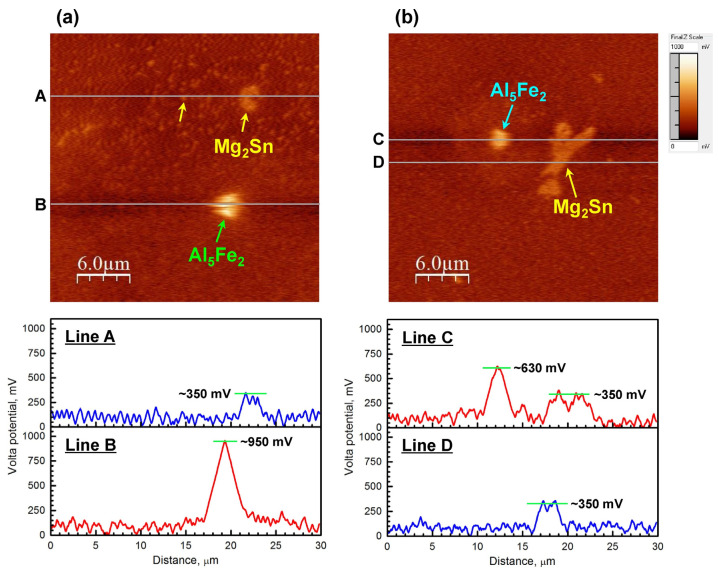
SKPFM diagram and volt potential distribution of Mg_2_Sn (yellow arrow) and Al_5_Fe_2_ (green arrow) of low-temperature extrusion (**a**) and high-temperature extrusion alloy (**b**) [88].

**Figure 6 materials-16-03791-f006:**
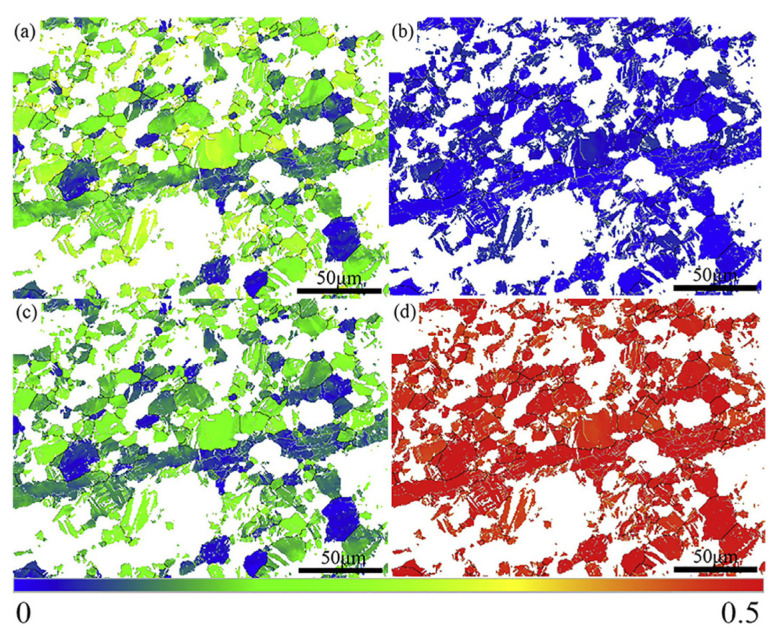
Schmid factor of (**a**) {0001}$<11\bar{2}0>$ basal slip, (**b**) $\{1\bar{1}00\}<11\bar{2}2>$ prismatic slip, (**c**) $\{1\bar{1}01\}<11\bar{2}0>$ first-order pyramidal slip, (**d**) $\{11\bar{2}2\}<\bar{1}\bar{1}23>$ second-order pyramidal slip [95].

**Figure 7 materials-16-03791-f007:**
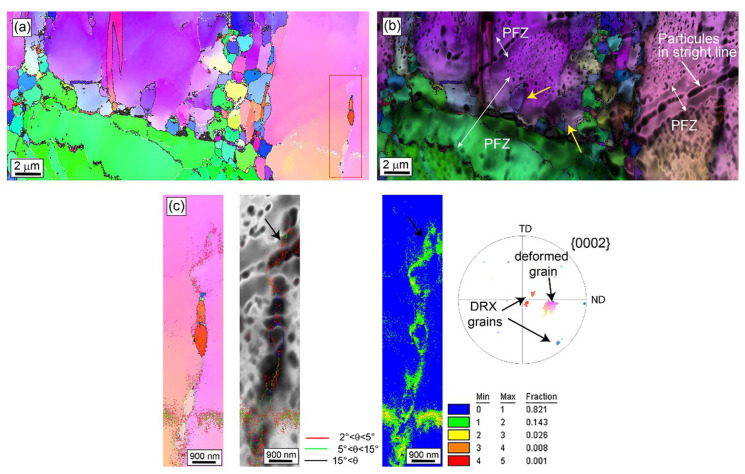
EBSD image of Mg-Gd-Zn-Zr alloy: (**a**) ED-IPF map, (**b**) IQ + ED-IPF map and (**c**) ED-IPF, IQ and KAM maps of the red box zone selected from (**a**) [104].

**Figure 8 materials-16-03791-f008:**
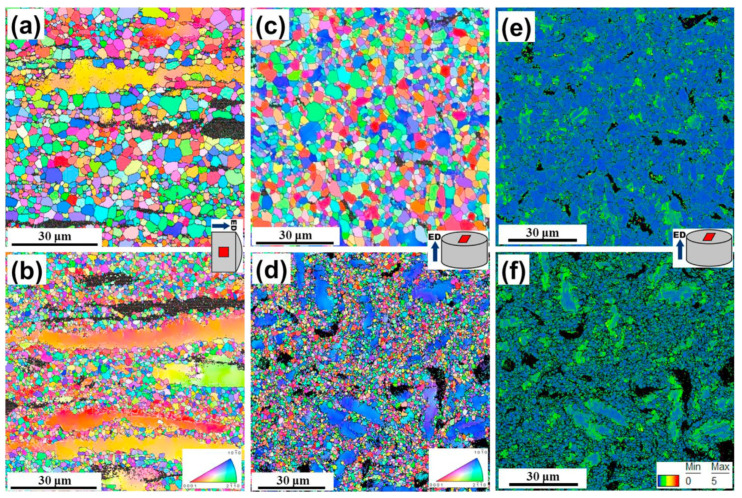
(**a**–**d**) EBSD results and KAM maps (**e**,**f**) of the as-extruded alloys obtained from (**a**,**b**) longitudinal and (**c**–**f**) transverse sections: (**a**,**c**,**e**) FE sample and (**b**,**d**,**f**) FHE sample [106].

**Table 1 materials-16-03791-t001:** Typical mechanical properties of extruded Mg alloys.

Compositions (wt.%)	Extrusion Parameters T, *v* (mm/s), ER	UTS (MPa)	TYS (MPa)	EL (%)	Ref.
Mg-3.07Al-0.78Zn-0.38Mn	430 °C, 20, 101:1	340.3	181.3	21.8	[45]
Mg-2.25Nd-0.11Zn-0.43Zr	290 °C, 50, 8:1	247 ± 4.4	204 ± 5.3	20.6 ± 1.6	[46]
Mg-9Gd-5Y-0.5Zr	350 °C, 7, 10:1	333	237	31.5	[18]
Mg-3Zn-1Y-1Mn	360 °C, 1, 16:1	243 ± 3	177 ± 2	19.6 ± 1.2	[47]
Mg-1Mn-0.5Al	300 °C, 333, 25:1	235	224	35.9	[48]
Mg-6Zn-1Y-0.85Zr	250 °C, 1, 16:1	410 ± 2.8	365 ± 3.0	7.1 ± 0.6	[49]
Mg-1.5Gd-1Zn	300 °C, 0.5, 25:1	316 ± 2	314 ± 1	21 ± 0.8	[50]
Mg-3.16Y-1.85Zn-0.37Zr	360 °C, 0.5, 9:1	330	280	21	[51]
Mg-2Zn-1.6Ca	400 °C, 3-4, 30:1	-	283.47	18.08	[20]
Mg-5Li-3Sn-2Al-1Zn	250 °C, 2	270	178	21.03	[4]
Mg-8Li-1Al-0.5Sn	260 °C, 25:1	322	240	11	[52]
Mg-0.5Gd	250 °C, 33.3, 25:1	301 ± 2	295 ± 2	14.9 ± 0.3	[53]
Mg-10Gd-3Y	400 °C, 22, 4:1	249.9 ± 4.8	198.1 ± 6.2	20.64 ± 3.3	[54]
Mg-4Sm-0.6Zn-0.4Zr	300 °C, 0.1, 6.25:1	435 ± 3	430 ± 2	6.0 ± 1.2	[55]
Mg-6Zn-0.5Zr	180 °C, 0.1, 7:1	392	370	13.9	[56]
Mg-7.5Gd-1.5Y-0.4Zr-0.5Ag	420 °C, 0.2, 9:1	329 ± 2	239 ± 2	15.6 ± 0.4	[57]
Mg-5.9Zn-1RE-0.6Zr	210 °C, 3, 9:1, 16:1	360	317	13.9	[58]
AZ91D	240 °C, 0.2, 28:1	382	320	13.8	[59]
Mg-6Gd-3Y-0.5Nd-0.5Zr	410 °C, 2, 10:1	279	192	24.9	[60]
Mg-6Zn-0.5Zr	390 °C, 1	313 ± 4.6	279 ± 4.7	12.1 ± 0.4	[28]
Mg-10Gd-3Sm-0.5Zr	300 °C, 5, 9.67:1	345	287	6.4	[61]
Mg-5Al-1Mn-0.5Zn-0.5Ca	360 °C, 16, 25:1	317 ± 1	247 ± 5	19.1 ± 0.8	[62]
Mg-9Gd-3Y-2Zn-0.4Zr	410 °C, 1.2	357	242	9	[63]
Mg-10Gd-2Y-0.5Zr	350 °C, 9.3:1	314	232	22.9	[64]
Mg-6.91Y-4.21Sm-0.60Zn-0.19Zr	350 °C, 0.75, 25:1	-	369	12	[65]
AZ31	400 °C, 0.55, 51:1	330 ± 5	216 ± 5	18.9 ± 0.4	[66]
Mg-13.2Gd-4.3Ni	400 °C	484	334	7.4	[67]
Mg-0.4Mn	150 °C, 0.1, 12:1	175	128	22	[68]
Mg-1Bi-1Mn-1Al-0.5Ca-0.3Zn	300 °C, 0.5, 20:1	438 ± 5.4	425 ± 6.8	2.1 ± 0.4	[69]
Mg-5.56Zn-1.79Yb-0.31Zr	320 °C, 0.2, 20:1	370	259	25.9	[34]
Mg-8Gd-4Y-0.5Mn-0.2Sc	200 °C, 0.1, 10:1	380	315	11	[70]
Mg-0.6Al-0.28Ca-0.25Mn	350 °C, 0.1, 12:1	232 ± 2.7	165 ± 7.8	14 ± 0.3	[71]
Mg-11Gd-3Y-0.5Nd-Zr	360 °C, 1, 16:1	334	256	14.9	[72]
Mg-8.2Gd-3.8Y-1.0Zn-0.4Zr	450 °C, 0.1, 20:1	394 ± 1	344 ± 1	21.5 ± 0.6	[73]
Mg-1.8Zn-1.74Gd-0.5Y-0.4Zr	340 °C, 6, 12:1	-	233.6	25	[74]
Mg-8Bi-1Al-1Zn	300 °C, 1.1, 30:1	331	291	14.6	[75]
Mg-1.11Sn-0.87Mn	300 °C, 1, 25:1	277	201	14.8	[76]
Mg-1Ca-1Mn	280 °C, 41.67, 25:1	322	305	18.2	[77]
Mg-2Zn-2Mn	280 °C, 41.67, 25:1	315	290	24	[78]
Mg-5Gd-3Y-1Zn-0.5Zr	400 °C, 1, 27:1	317 ± 5.3	238 ± 0.8	7.7 ± 0.9	[79]
Mg-10.22Li-4.73Zn-0.42Er	150 °C, 25:1	272	271	53.1	[80]

## Data Availability

Not applicable.

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
