# Peer review of "Research Progress on Microstructure Evolution and Strengthening-Toughening Mechanism of Mg Alloys by Extrusion"

_materials, 2023, doi:10.3390/ma16103791_

Round 1

Reviewer 1 Report

The review paper is well-organized and clearly written. The summary of the effect of extrusion is comprehensive and in-depth.

Two minor recommendations: 

1. The review of rotary extrusion is thin and only in one fashion. I recommend adding discussions about other rotary extrusion processes: e.g., friction extrusion, Shear assisted extrusion and processing, screw extrusion, etc. 

2. As a reader of a review paper, I expect authors to provide some forward-thinking ideas and advice in the Mg extrusion field with standing on the outstanding summary.

Author Response

Response to Reviewer Comments

We thank the Reviewer for reading our manuscript carefully and give positive comments. We have carefully considered all comments from the reviewers and revised our manuscript accordingly. The manuscript has also been double-checked, and the typos and grammar errors we found have been corrected. In the following section, we summarize our responses to each comment from the reviewers. We believe that our responses have well addressed all concerns from the reviewers. We hope our revised manuscript can be accepted for publication.

  1. The review of rotary extrusion is thin and only in one fashion. I recommend adding discussions about other rotary extrusion processes: e.g., friction extrusion, Shear assisted extrusion and processing, screw extrusion, etc.

Response: Thank you for pointing out weakness in our manuscript. We added friction stir extrusion and shear assisted processing and extrusion – ‘Li et al [30] proposed another rotary extrusion process named friction stir extrusion and increased the YS, UTS and compression strain of rare-containing Mg alloys by 42.5%, 63.6% and 35.5%, respectively. Shear assisted processing and extrusion (ShAPE) could work with more than 20 times less ram force compared to conventional extrusion [31]. The reduction is mainly due to the rotational shear component applied by ShAPE interacting with scrolled die face, which promotes the flow of Mg alloy into the die orifice’.

  1. As a reader of a review paper, I expect authors to provide some forward-thinking ideas and advice in the Mg extrusion field with standing on the outstanding summary.

Response: Thank you for helpful advice to our manuscript. We add ‘5.       Due to the significant enhancement effect of extrusion process on magnesium alloys, the extrusion pressure should be reduced in the future by combining the optimization of extrusion process with the setting of die parameters.’ in Conclusion section (Line 628).

Reviewer 2 Report

Errors:
-          Line 39 – “tensile yield strength (TYS)” please correct to:  tensile yield strength (YS)

-          Line 83 – “2. Magnesium alloy extrusion process and advantages” advantages of what ? please clarify or rewrite the subtitle.

-          Line 95 – „backward extrusion, reverse extrusion” please change to: direct and indirect extrusion.

-          Line 98 – “Forward extrusion” please change to: Direct extrusion (please correct this along the text in the article)

-          Line 127 – „2.2 Backward extrusion” please change to Indirect extrusion (please correct this along the text in the article)

-          Line 226 – “TS” please correct to: UTS

General remarks on the article:

-          It is a review article and due to the above the title of article is somehow misleading for me. Maybe it would be beneficial to change it a little bit in order to make it more clear and more related to the article characteristic.

-          As for a review article the bibliography is not the strongest part of the work. It is mostly oriented to one region of the world, however the subject is very somehow very popular all around the world so there are plenty other choices to cite in the work.

In general English along the paper is good. No major errors were found.

Author Response

Response to Reviewer Comments

We thank the Reviewer for reading our manuscript carefully and give positive comments. We have carefully considered all comments from the reviewers and revised our manuscript accordingly. The manuscript has also been double-checked, and the typos and grammar errors we found have been corrected. In the following section, we summarize our responses to each comment from the reviewers. We believe that our responses have well addressed all concerns from the reviewers. We hope our revised manuscript can be accepted for publication.

  1. Line 39 – “tensile yield strength (TYS)” please correct to: tensile yield strength (YS)

Response: Thank you for pointing out this problem in our manuscript and we feel so sorry for the confusion brought to reviewer and readers. According to the revised content, ‘tensile yield strength (TYS)’ has changed to ‘tensile yield strength (YS)’.

  1. Magnesium alloy extrusion process and advantages” advantages of what ? please clarify or rewrite the subtitle

Response: We are very sorry for the misunderstanding problem. According to the advice, ‘Magnesium alloy extrusion process and advantages’ has changed to ‘Magnesium alloy extrusion process’.

  1. backward extrusion, reverse extrusion” please change to: direct and indirect extrusion

Response: Thank you for your advice. The forward extrusion and back extrusion in this paper are classified according to the metal flow direction. The direct and indirect extrusion are classified according to the pressure application mode, which is not well suitable in this article.

  1. Forward extrusion” please change to: Direct extrusion (please correct this along the text in the article

Response: Thank you for your advice. The forward extrusion in this paper depends on the metal flow direction instead of the pressure application mode and it might be more suitable in this article.

  1. 2.2 Backward extrusion” please change to Indirect extrusion (please correct this along the text in the article)

Response: Thank you for your advice. The backward extrusion in this paper is due to the metal flow direction, which might be more suitable in this article.

  1. Line 226 – “TS” please correct to: UTS

Response: We are very sorry for the error. According to the advice, ‘TS’ has changed to ‘UTS’.

  1. It is a review article and due to the above the title of article is somehow misleading for me. Maybe it would be beneficial to change it a little bit in order to make it more clear and more related to the article characteristic.

Response: Thank you for your advice. The title has changed to ‘Research progress on microstructure evolution and strengthening-toughening mechanism of Mg alloys by extrusion’.

  1. As for a review article the bibliography is not the strongest part of the work. It is mostly oriented to one region of the world, however the subject is very somehow very popular all around the world so there are plenty other choices to cite in the work.

Response: Thank you for your advice. We will pay more attention to this problem in the future research.

Reviewer 3 Report

The article is very well written and presents a comprehensive summary of papers in specialty literature, with an excellent description of the importance of magnesium-based alloys in various sectors, due to their special characteristics.

This paper presents the particular importance constitutes the extrusion process often used to eliminate the structural defects of magnesium alloys, improve strength and toughness synergy, and corrosion resistance, etc. The technical observations are necessary for establishing the importance of this process for improving the deformation capacity of the alloy expanding the application range of magnesium alloys.

            The introduction provides a good, generalized background of the topic, making a detailed description regarding the relevance of the main advanced processes (such as die-casting, rolling and forging, rotary extrusions, etc.) used for magnesium alloys processing, confirmed with the extensive bibliography from the last years.

The research progress regarding the influence of the extrusion process on the microstructure, mechanical properties, and corrosion fatigue performance of magnesium alloys, including the relationship between deformation and strengthening mechanism of extrusion are detailed.   

The conclusions are clearly formulated and justified.

Author Response

Response to Reviewer Comments

We thank the Reviewer for reading our manuscript carefully and give positive comments. We have carefully considered all comments from the reviewers and revised our manuscript accordingly. The manuscript has also been double-checked, and the typos and grammar errors we found have been corrected. In the following section, we summarize our responses to each comment from the reviewers. We believe that our responses have well addressed all concerns from the reviewers. We hope our revised manuscript can be accepted for publication.

The article is very well written and presents a comprehensive summary of papers in specialty literature, with an excellent description of the importance of magnesium-based alloys in various sectors, due to their special characteristics.

This paper presents the particular importance constitutes the extrusion process often used to eliminate the structural defects of magnesium alloys, improve strength and toughness synergy, and corrosion resistance, etc. The technical observations are necessary for establishing the importance of this process for improving the deformation capacity of the alloy expanding the application range of magnesium alloys.

            The introduction provides a good, generalized background of the topic, making a detailed description regarding the relevance of the main advanced processes (such as die-casting, rolling and forging, rotary extrusions, etc.) used for magnesium alloys processing, confirmed with the extensive bibliography from the last years.

The research progress regarding the influence of the extrusion process on the microstructure, mechanical properties, and corrosion fatigue performance of magnesium alloys, including the relationship between deformation and strengthening mechanism of extrusion are detailed.   

The conclusions are clearly formulated and justified.

Response: Thank you for your affirmation. The article has been optimized for language.

Reviewer 4 Report

Line 18: probably extrusion processes ?

Line 204: Change in “Effect of extrusion on Mg-alloys

Line 258: It is suggested to add here a table with main tensile properties and references.

The paper is well written, some minor details to be checked

Author Response

Response to Reviewer Comments

We thank the Reviewer for reading our manuscript carefully and give positive comments. We have carefully considered all comments from the reviewers and revised our manuscript accordingly. The manuscript has also been double-checked, and the typos and grammar errors we found have been corrected. In the following section, we summarize our responses to each comment from the reviewers. We believe that our responses have well addressed all concerns from the reviewers. We hope our revised manuscript can be accepted for publication.

  1. Line 18: probably extrusion processes ?

Response: Thank you for pointing out this problem in our manuscript and we feel so sorry for the confusion brought to reviewer and readers. According to the revised content, ‘extrusion process’ has changed to ‘extrusion processes’.

  1. Line 204: Change in “Effect of extrusion on Mg-alloys

Response: We are very sorry for our error. According to the revised content, ‘Effect of extrusion on alloys’ has changed to ‘Effect of extrusion on Mg-alloys’.

  1. It is suggested to add here a table with main tensile properties and references

Response: Thank you for pointing out this problem in our manuscript. A table with tensile properties and reference has been added.

Reviewer 5 Report

Review article "Research progress on microstructure evolution and strengthening-toughening mechanism of extruded magnesium alloys" is devoted to the actual topic. Processing methods for magnesium alloys are indeed relevant in various industries. The review covers a wide range of deformation processes. The structure of the review article is logical. The quality of the figures is good.

However, there are a couple of issues that should be clarified and supplemented in the article:

1. Only 2 types of products are considered in the work? Round rods (Fig. 1 and Fig. 3) and wheels (Fig. 2 and Fig. 4)? Were other profiles considered by the authors?

2. With regard to rods of small cross-section, it would be advisable to mention the methods of continuous extrusion in which the lack of discreteness is eliminated.

I recommend accepting the article for publication after these minor revision.

Author Response

Response to Reviewer Comments

We thank the Reviewer for reading our manuscript carefully and give positive comments. We have carefully considered all comments from the reviewers and revised our manuscript accordingly. The manuscript has also been double-checked, and the typos and grammar errors we found have been corrected. In the following section, we summarize our responses to each comment from the reviewers. We believe that our responses have well addressed all concerns from the reviewers. We hope our revised manuscript can be accepted for publication.

  1. Only 2 types of products are considered in the work? Round rods (Fig. 1 and Fig. 3) and wheels (Fig. 2 and Fig. 4)? Were other profiles considered by the authors?

Response: Thank you for your advice. The reason can be listed as follow: Till now, the extrusion products are mainly composed of rods, pipes and plates, etc. However, the representativeness of products with complex structure is low. Therefore, considering the current product application range and scene, the paper involves the mainstream products.

  1. With regard to rods of small cross-section, it would be advisable to mention the methods of continuous extrusion in which the lack of discreteness is eliminated.

Response: Thank you for your advice. The content has been optimized according to the requirements.
